# Transformers trained on proteins can learn to attend to Euclidean distance

## Abstract

While conventional Transformers generally operate on sequence data, they can be used in conjunction with structure models, typically SE(3)-invariant or equivariant graph neural networks (GNNs), for 3D applications such as protein structure modelling. These hybrids typically involve either (1) preprocessing/tokenizing structural features as input for Transformers or (2) taking Transformer embeddings and processing them within a structural representation. However, there is evidence that Transformers can learn to process structural information on their own, such as the AlphaFold3 structural diffusion model. In this work we show that Transformers can function independently as structure models when passed linear embeddings of coordinates. We first provide a theoretical explanation for how Transformers can learn to filter attention as a 3D Gaussian with learned variance. We then validate this theory using both simulated 3D points and in the context of masked token prediction for proteins. Finally, we show that pre-training protein Transformer encoders with structure improves performance on a downstream task, yielding better performance than custom structural models. Together, this work provides a basis for using standard Transformers as hybrid structure-language models.

## 1 Introduction

### 1.1 Background

Transformers typically operate on sequential data, however many applications of Transformers benefit from an ability to learn geometric reasoning. For instance, ESM-2 (Lin et al., 2023) demonstrates that in order to effectively predict masked tokens in protein *sequences*, the model has learned some ability to predict protein *structures*. Other tasks such as image processing or even natural language processing may benefit from an internal representation of objects in 3D space. However, it is unclear how Transformers can learn to use 3D representations to perform spatial reasoning. To this end, custom structural Transformers have been created which model data as graphs and represent distance between nodes as edge features in order to perform SE(3)-invariant attention (Ingraham et al., 2019; Fuchs et al., 2020; Liao & Smidt, 2023; Liao et al., 2023).

SE(3) invariance means that all functions of coordinates reduce to functions of relative distance. That is, for every function $f$ of two coordinates $x_1$ and $x_2$, there exists an equivalent function $g$ which depends only on their relative distance: $f(\vec{x_1}, \vec{x_2}) = g(|\vec{x_1} - \vec{x_2}|)$. Since relative distance in Euclidean space is defined as $|\vec{x}| = \sqrt{x^2 + y^2 + z^2}$, it may be easier to learn functions of the square of the relative distance — a linear combination of functions of the individual coordinates.

In this manuscript, we investigate how conventional Transformers can learn to approximate functions of the squared distance between points, thereby learning an approximately SE(3)-invariant measure of distance. In short, we show that "out of the box" Transformers can act as 3D structural models. Our main contributions are (1) to provide a theoretical explanation for how standard Transformers can learn to measure distance and perform structural reasoning, (2) to show that Transformers indeed learn Gaussian functions of distance and investigate efficient data augmentation methods which can be used to learn SE(3), and (3) to train a protein masked token prediction model with coordinates and show that finetuning it for function prediction yields a model which outperforms structural GNNs.

## 1.2 PRIOR WORK

There have been prior approaches to merge Transformers with SE(3)-(in/equi)variant models, especially for computational chemistry and 3D point clouds. Some methods add attention blocks to SE(3) GNNs to create SE(3)-invariant GNN Transformers (Fuchs et al., 2020; Liao & Smidt, 2023). These have shown good results on a number of tasks, however tend to be memory-intensive, particularly because attention is performed on edges, which grow as $n^2$ for fully-connected graphs. As a result, the graph connectedness of the GNNs is typically limited to k-nearest neighbours. In contrast, memory-efficient attention implementations such as FlashAttention (Dao et al., 2022; Dao, 2023) have enabled linear-memory standard Transformers.

Previous works have demonstrated that sequence-only protein Transformers can learn attention maps which correlate with physical contacts (Lin et al., 2023; Vig et al., 2020). However, these works do not formally model structure and so are limited by which contacts can be predicted from purely sequential patterns. To overcome this, Transformers are often paired with structural models, for instance methods such as ProSST (Li et al., 2024), ESM-IF (Hsu et al., 2022), and ESM3 (Hayes et al., 2024) use custom graph-based modules to create structural tokens which are fed into standard Transformers. In contrast, our work shows that standard Transformers are natively capable of using coordinates to model structure and measure distance.

Similarly, AlphaFold2 (Jumper et al., 2021) and ESMFold (Lin et al., 2023) use Transformers to preprocess protein sequences for structure prediction. Again, these preprocessed representations have been shown to correlate with structural contacts. AlphaFold2 makes this explicit during training by minimizing a distrogram loss which encourages the EvoFormer to learn structural contacts. However, it is unclear if these representations are learning to explicitly embed coordinates in 3D and both models still require an SE(3)-equivariant GNN structure module to actually produce 3D structures.

In building AlphaFold3, DeepMind replaced AlphaFold2's SE(3)-equivariant structure module with linearly embedded coordinates fed into a diffusion transformer (Abramson et al., 2024). AlphaFold3's structure module uses inner product attention with a pair bias learned from the pair representation. At present, this still requires quadratic memory, however does indicate that nearly-standard Transformers with linearly embedded coordinates can learn on structure. Here, we explore how such linearly embedded coordinates can be used by standard Transformer attention modules to measure the Euclidean distance between tokens, and, in contrast to prior work, show that no modifications are necessary for the standard Transformer architecture to learn to perform structural reasoning.

## 2 THEORY

Here we introduce our theory for how Transformers can learn to measure Euclidean distance. Throughout this section, we assume that all coordinates are small. This can be achieved by scaling the inputs and outputs manually or via learned weights in the linear input/output maps. For more detail, see Appendix A.1.

### 2.1 GAUSSIAN SPATIAL ATTENTION

Consider a pre-norm Transformer (Xiong et al., 2020) operating on a sequence of embedded positions, $E(\vec{x_i})$. For simplicity, assume that the key and query embeddings for all heads are trivial ($Q = K = I_d$). Such mappings can easily be learned as long as the head dimension is at least $d$. Then, in the first layer of the Transformer, the attention matrix for all heads will be:

$$A_{i,j} = SM(\frac{LN(E(\vec{x_i})) \cdot LN(E(\vec{x_j}))}{\sqrt{d}})$$ (1)

Where SM denotes the softmax function and LN denotes LayerNorm (Ba et al., 2016).

Our objective is to determine an embedding, $E$, such that the attention $A_{i,j}$ is a monotone decreasing function of the Euclidean distance between $\vec{x_i}$ and $\vec{x_j}$. In particular, if we choose $E$ such that, for some $a, b \in \mathbb{R}$:

$$LN(E(\vec{x_i})) \cdot LN(E(\vec{x_j})) \approx -a|\vec{x_i} - \vec{x_j}|^2 + b \qquad (2)$$

Then,

$$
\begin{aligned}
A_{i,j} &= SM(\frac{LN(E(\vec{x_i})) \cdot LN(E(\vec{x_j}))}{\sqrt{d}}) \\
&\approx SM(-\frac{a}{\sqrt{d}}|\vec{x_i} - \vec{x_j}|^2 + \frac{b}{\sqrt{d}}) \\
&= SM(-\frac{a}{\sqrt{d}}|\vec{x_i} - \vec{x_j}|^2)
\end{aligned}
\qquad (3)
$$

In particular, the unnormalized attention paid to $x_j$ by $x_i$ is:

$$A_{nonorm_{i,j}} \approx e^{-\frac{a}{\sqrt{d}}|\vec{x_i} - \vec{x_j}|^2} \qquad (4)$$

Which is a Gaussian of the relative distance between the two points, i.e., Transformers can learn to approximate a 3D Gaussian to gate attention values and selectively attend to points nearby in 3D space. By learning individual LayerNorm gains or $Q/K$ mappings, each head can tune the variance of this Gaussian filter, which allows each head to determine the appropriate spatial resolution for that type of information.

Below, we provide embeddings which satisfy Equation 2 and so lead to Gaussian attention filters.

## 2.2 SPATIAL POSITIONAL EMBEDDINGS

For simplicity, we consider the case of 1 spatial dimension and provide 4-dimensional embeddings, which satisfy Equation 2. The 3 (or $n$) dimensional case is similar. Consider the embeddings:

$$
\begin{aligned}
E_{trig}(x) &= (\sin(x), -\sin(x), \cos(x), -\cos(x)) \\
E_{lin}(x) &= (x, -x, 1, -1) \\
E_{quad}(x) &= (x, -x, 1 - \frac{x^2}{2}, \frac{x^2}{2} - 1)
\end{aligned}
\qquad (5)
$$

Here, $E_{trig}$ is similar to the standard sinusoidal positional encoding for linear sequences (Vaswani et al., 2017). Then, $E_{lin}$ and $E_{quad}$ can be thought of as the first and second order approximations of $E_{trig}$, respectively. It can be shown (see Appendix A.2) that for all three embeddings:

$$LN(E(x_i)) \cdot LN(E(x_j)) \approx -2|x_i - x_j|^2 + 4 \qquad (6)$$

Subject to $|x_i - x_j|$ being small for $E_{trig}$ and subject to $|x_i|$ and $|x_j|$ being small for $E_{lin}$ and $E_{quad}$. This surprising result for $E_{lin}$ stems from the non-linearity of LayerNorm causing the constant terms to be locally quadratic. For more details, see Appendix A.2.2. An important consequence of this result is that simple linear embeddings of positions can still lead to an approximately 3D Gaussian filter of relative distance for attention.

Additionally, the approximation $E_{quad}$ is better than that of $E_{lin}$, however the requirement to encode $x^2$ explicitly makes it harder for individual attention heads to rescale the positions appropriately, since heads have to learn the linear and quadratic scaling terms separately. However, in A.2 we show that ReGLU and SwiGLU activation functions (Shazeer, 2020) are capable of learning exactly quadratic functions of their input, which may allow some modern Transformers to learn positional embeddings of the form $E_{quad}$ after the appropriate rescaling. This may be a useful benefit of GLU activation functions for models such as AlphaFold3 (Abramson et al., 2024).

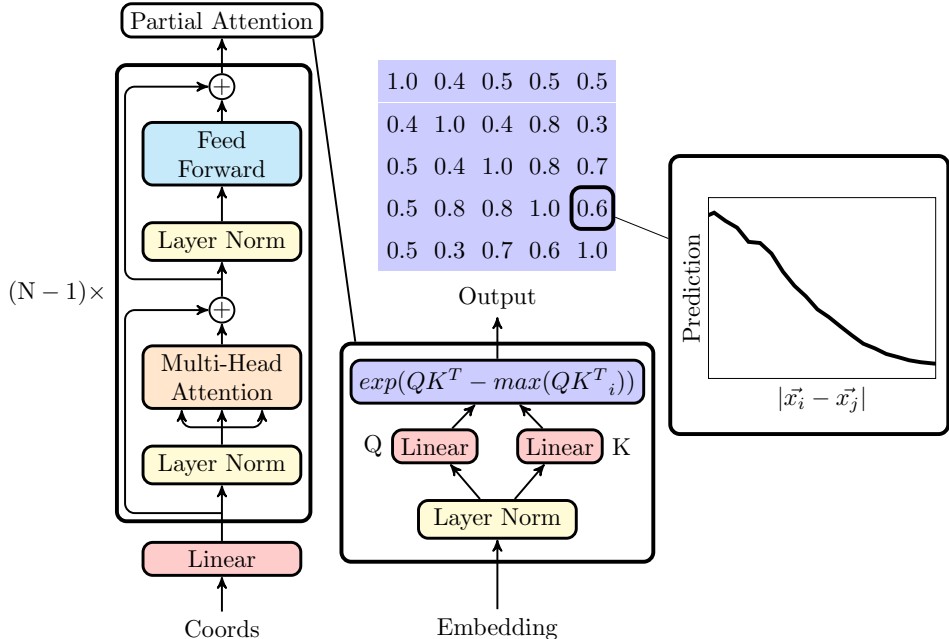

Figure 1: Overview of the simulated experiment model. Coordinates are passed through a Transformer encoder which is truncated such that the output is the unnormalized attention for a single head. Loss is computed as the difference between the attention for each pair and a Gaussian of the relative distance between those points.

## 3 EXPERIMENTS

### 3.1 SIMULATED POINTS

To test our theory of how Transformers learn to measure distance, we designed a Transformer encoder which is truncated such that the output is the unnormalized attention matrix for a single head. A diagram of this model is shown in Figure 1. We computed the loss as the $l_1$ difference between the output matrix and the matrix $A_{i,j} = e^{\left(\frac{-(x_i - x_j)^2}{200^2}\right)}$. This corresponds to the prenormalized softmax of the negative square of the relative distance between points, as predicted by our theory. The data consisted of 10,000 "structures", each with five 3-dimensional points with coordinates randomly selected between 0 and 200. Unless otherwise indicated, the Transformer encoder for all experiments has three layers (is truncated at the third layer), an embedding dimension of 256, a feedforward dimension of 1,024, 8 heads, pre-normalization and ReLU activation. The models were trained with a batch size of 16 using the Adam optimizer with a peak learning rate of $4 \times 10^{-4}$ which is reached after 4,000 warmup steps, and then is quadratically decayed.

As in AlphaFold3 (Abramson et al., 2024), we transform the input structures before they are passed through the model. Whenever a structure is loaded, its points are recentred, randomly rotated, and rescaled by a factor of $\frac{1}{16}$. This has two benefits. First, recentering and rescaling the points ensures that all coordinates stay relatively small, even before the embedding layer has learned an appropriate mapping. Second, recentering and randomly rotating gives the model resilience to translations and rotations which encourages it to learn a distance measure which is truly SE(3)-invariant.

### 3.1.1 TRANSFORMERS CAN ATTEND TO $|\vec{x_i} - \vec{x_j}|^2$

In Section 2, we show that Transformers are theoretically capable of learning Gaussian functions of distance. We experimented with learning $e^{-|\vec{x_i} - \vec{x_j}|^p}$ for different powers $p$, ranging from 0.5 to 4, in increments of 0.5. Figure 2a shows the relationship between $p$ and the validation loss. As expected, Transformers can learn to reproduce the $e^{-|\vec{x_i} - \vec{x_j}|^2}$ attention matrix most accurately which shows that a Gaussian is the most natural way for Transformers to learn to filter attention spatially.

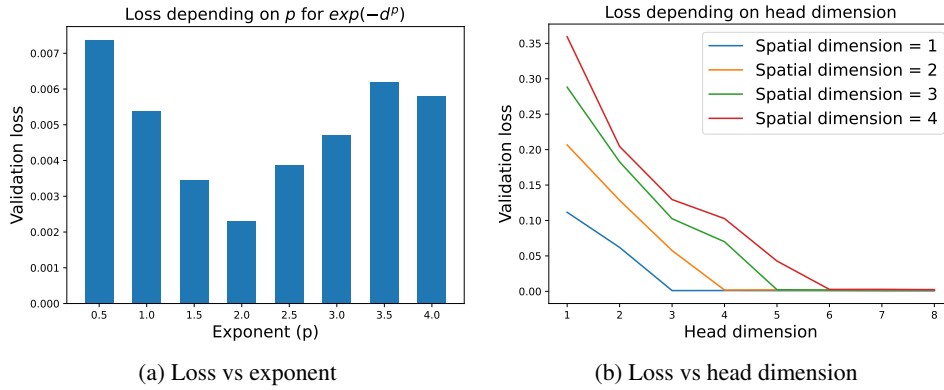

(a) Loss vs exponent          (b) Loss vs head dimension

Figure 2: (a) Validation loss as a function of the exponent $p$. The loss is lowest for $p = 2$ which corresponds to learning a Gaussian function of distance. (b) Validation loss as a function of head dimension for different spatial dimensions. Models need a head dimension of $n + 2$ to accurately measure distance in $\mathbb{R}^n$.

### 3.1.2 TRANSFORMERS NEED $n + 2$ EMBEDDING DIMENSIONS TO LEARN DISTANCE IN $\mathbb{R}^n$

Next, we explored how large a spatial embedding must be to learn a good approximation of distance for $\mathbb{R}^n$. For the case of 1 spatial dimension, we provide a theoretical 4-dimensional embedding. We trained models to predict Euclidean distance in $\mathbb{R}^n$ for $n \in 1, 2, 3, 4$ using head dimensions from 1 to 8. Figure 2b shows the relationship between validation loss and head dimension. We found that the model only needs $n + 2$ head dimensions to learn a good approximation of distance for all spatial dimensions. This means that to learn a good approximation of distance in $\mathbb{R}^3$, a model must reserve at least 5 embedding dimensions for each head. This is a surprisingly compact requirement which should be easily accommodated even in small Transformers.

### 3.1.3 SE(3) TRANSFORMATIONS IMPROVE LEARNED SE(3)-INVARIANCE

We investigated whether Transformers will learn to overfit training data in a low data regime and if this can be prevented. This could also correspond to a scenario where there is only a strong structural signal in a small number of training examples. We reduced the number of training points to 100 and measured the training and validation loss. To test the importance of data augmentation, we trained models with and without the random rotations (Figure 3a). The raw coordinates clearly demonstrate overfitting while the randomly rotated coordinates show near perfect alignment between training and validation loss. Importantly, this form of data augmentation does not require creating new data points, only rotating the training data each epoch.

We measured the average $\ell_1$ distance between predictions of randomly rotated structures for the models trained with and without random rotations, as a measure of SE(3) divergence (Figure 3b). In both cases the SE(3) divergence was almost the same as the validation loss, indicating that randomly rotating training structures reduces overfitting by encouraging models to learn an SE(3)-invariant measure of distance.

### 3.2 PROTEINS

Proteins are a natural fit for structural Transformers because they are composed of linear sequences embedded in 3D space. As such, their properties depend on both sequential and structural features. We considered two tasks in protein modelling: predicting masked tokens as a pretraining objective and predicting protein function conditioned on embeddings generated by pretrained models. For all protein experiments we used the GO PDB dataset from DeepFRI (Gligorijević et al., 2021) which comprises ∼36K protein chains.

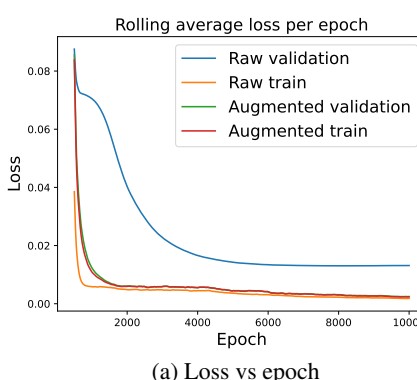
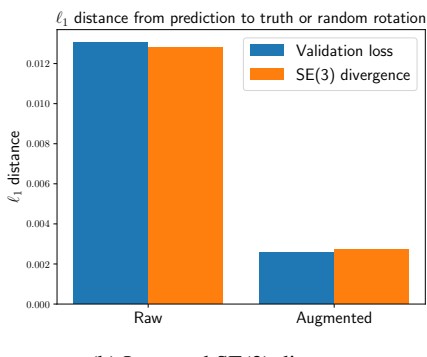

(a) Loss vs epoch

(b) Loss and SE(3) divergence

Figure 3: (a) Training and validation loss per epoch in a low data setting. The rolling average of the last 500 epochs is shown. Raw coordinates show large divergence between training and validation loss whereas randomly rotated coordinates show near perfect alignment, resulting in a substantially lower validation loss. (b) The final validation loss is very close to the average predicted distance between randomly rotated structures, indicating that the validation loss is minimized by learning a more SE(3)-invariant measure of distance.

### 3.2.1 PRETRAINING A STRUCTURAL PROTEIN LANGUAGE MODEL

To test a Transformer's ability to learn useful structural patterns in proteins, we trained an ESM/BERT-style (Rives et al., 2021; Devlin et al., 2019) model to complete masked token prediction. We trained two models: one with coordinates ('coords model') and the other without ('non-coords model'). The version with coordinates added a linear embedding of the coordinates to the token embedding in the same way as the simulated experiments. Both models were very similar to the smallest publicly released ESM1 model, consisting of a 6-layer Transformer encoder with a hidden dimension of 768, 12 attention heads, a feedforward dimension of 2048, and GeLU activation (Hendrycks & Gimpel, 2023). As in ESM, we omitted dropout. To prevent the model from focusing too much on linear positional information, we used Sinusoidal Positional Encodings rather than Rotary Positional Encodings (Su et al., 2022). As is common in masked token prediction, we masked 15% of tokens. Of the masked tokens, 80% were replaced with a [MASK] token, 10% were replaced with a random amino acid, and 10% were left unchanged. We clustered the data by 50% sequence identity using MMSeqs2 (Steinegger & Söding, 2017) and randomly held out 1% of the clusters to use as a validation set. We trained each model for 100 epochs with a fixed batch size of 24, resulting in approximately 150K updates. We used the Adam optimizer with 4,000 warmup steps to a peak learning rate of $2.3 \times 10^{-4}$, followed by inverse square decay. Each time a structure was loaded, its coordinates were recentred, randomly rotated, and rescaled.

As shown in Figure 4a, adding coordinates substantially improved the model, leading to a final training perplexity of 6.5 with coordinates vs 11.9 without. The final training loss for the version without coordinates (after 100 epochs) was surpassed by the version with coordinates after 8 epochs. Additionally, the final validation loss was surpassed after only 4 epochs which may indicate that the structural features learned early in training are more robust to dissimilarity in sequence space.

We also investigated the difference in sequence recovery rates between the two models. The total sequence recovery rate was ~23% for the non-coords model compared to ~38% for the coords model. Figure 4b shows a breakdown of the recovery rates per amino acid type. The recovery rate for the coords model was greater than or equal to that of the non-coords model for all amino acid types. The difference was particularly stark for glycine and proline, which may be related to their distinct backbone conformational preferences (Ho & Brasseur, 2005; Beck et al., 2008).

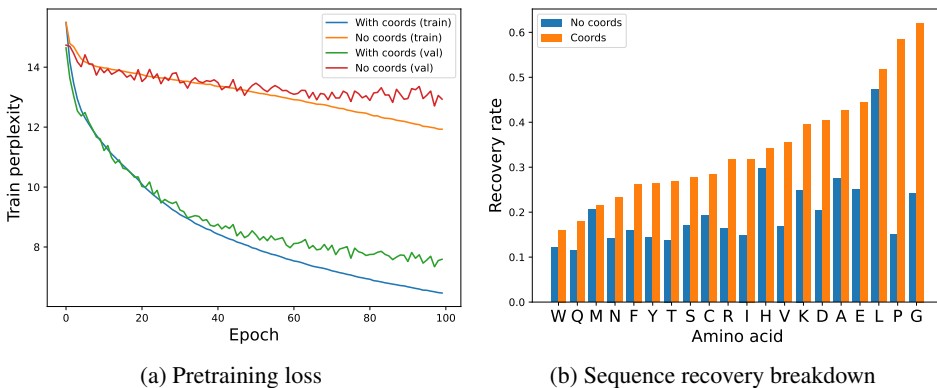

(a) Pretraining loss          (b) Sequence recovery breakdown

Figure 4: (a) Masked token prediction loss per epoch. Adding coordinates to the model hugely improves training and validation accuracy, as the model quickly learns to use coordinates to measure distance and compute structural features. (b) Sequence recovery rate for each amino acid type. The height of each bar represents the recovery rate for the coords model and the orange portion is the recovery rate for the non-coords model (the coords model had at least as high a recovery rate as the non-coords model for all amino acid types). The recovery rates for glycine and proline were substantially higher in the coords model, which may correspond to identification of beta turns.

### 3.2.2 PRETRAINED MODELS LEARN TO MEASURE DISTANCE

In the model trained in the previous paragraph, there are three inputs to the model for each token: amino acid type, sequential position, and 3D coordinates. To test if the pretrained model with coordinates was learning to measure distance as predicted, we plotted the average attention paid by each pair of tokens across all heads in a layer as a function of distance. We also plotted the average attention paid to each token as a function of relative sequence distance. To isolate the effect of each feature, we fixed all amino acids to alanine. We also fixed the linear sequence index to a constant value for all tokens while measuring 3D dependence and fixed the 3D position to $(0, 0, 0)$ while measuring linear dependence. For the distance measurements, we rounded each pairwise distance to the nearest Angstrom and computed the average for each distance value.

Figure 5 shows the plots for 3D and linear positional dependence for all layers for both models as well as the amplitude and standard deviation of the fit Gaussian for each layer. As expected, the model without coordinates shows no correlation with Euclidean distance. Conversely, the model with coordinates shows a strong dependence in the early layers which is well-approximated by a Gaussian. Both models progressively widen their field of view as information passes through the layers. This may correspond to a system of reasoning where the model collects information about the local environment before processing this information alongside global patterns. We provide a similar plot without isolating each of the factors in Appendix A.3.

### 3.2.3 PROTEIN FUNCTION PREDICTION

Finally, we tested whether the pretrained protein model embeddings could improve accuracy on a downstream task. We trained models to predict protein molecular function Gene Ontology labels (Ashburner et al., 2000). Protein function prediction has been studied extensively, and has been shown to benefit from both language and structural features (Gligorijević et al., 2021). We trained models to predict protein function based on the mean token embedding output by the pretrained models, using the same data splits as DeepFRI (Gligorijević et al., 2021). We compare our results to the DeepFRI and DeepCNN versions which were trained on PDB sequences. DeepFRI provided a useful comparison since it achieved state of the art performance on protein function prediction before models were trained using ESM embeddings. Thus we could pretrain our model on the same data as DeepFRI and evaluate the specific contributions of model architectures. DeepFRI is an ensemble of Graph Convolutional Networks (Kipf & Welling, 2022) with different propagation rules, as in Dehmamy et al. (2019). DeepFRI uses a pretrained LSTM model (Graves, 2014) to generate

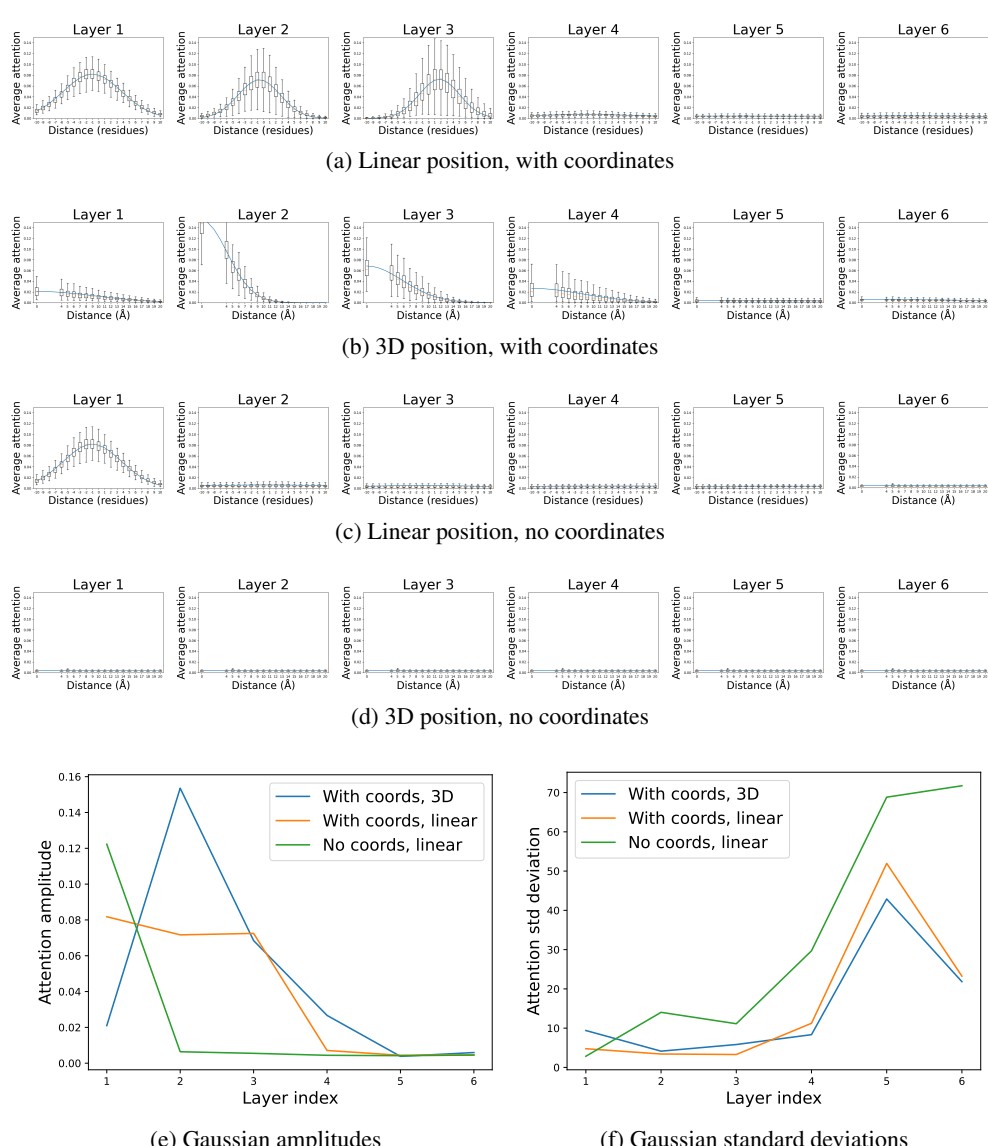

Figure 5: Average attention paid per layer to linear and 3D positional information (a-d). Fit Gaussian functions are shown in blue for each plot. The model which was trained with coordinates learns to filter heavily by linear and 3D positional information whereas the model trained without coordinates only filters by linear information in early layers. Amplitudes (e) and standard deviations (f) of the fit Gaussians show that both models pay attention to local features early and then gradually widen their fields of view.

language model embeddings for node features. DeepCNN is a Convolutional Neural Network (Le-Cun et al.) meant to replicate DeepGO (Kulmanov et al., 2018) but retrained on the same sequences as DeepFRI (and our model).

We train two models based on the pretrained models from the previous section. Our first model is based on the simple MLP model from Kulmanov et al. (2024), which is a simple 2 layer MLP block of 1,024 dimensions with a residual connection. The input to the model is a learned linear embedding of the mean token embedding output by the pretrained models.

Our second model is a finetuned version of the pretrained masked token prediction models. As in BERT (Devlin et al., 2019) the output is a learned linear projection of the final start token embedding. Each model (coords/no coords) is finetuned for 20 epochs with a constant learning rate of $3 \times 10^{-5}$.

Table 1: GO molecular function prediction results.

| Pretraining (# seqs) | Method | Structure | AUPRC | (Gain from structure) | Max F1 | (Gain from structure) |
|---|---|---|---|---|---|---|
| DeepFRI (∼10M) | DeepFRI | ✗ | 0.427 | | 0.438 | |
| | | ✓ | 0.446 | 0.019 | 0.460 | 0.022 |
| None | DeepCNN | ✗ | 0.363 | | 0.385 | |
| Ours (∼35K) | MLP | ✗ | 0.361 | | 0.377 | |
| | | ✓ | 0.460 | 0.099 | 0.465 | 0.088 |
| | Finetuned | ✗ | 0.381 | | 0.421 | |
| | | ✓ | 0.566 | 0.185 | 0.575 | 0.154 |

The results are shown in Table 1. Additional results for predicting biological process and cellular component are provided in Appendix A.4. Our sequence-only MLP model compared competitively with DeepCNN and our sequence-structure MLP model compared competitively with DeepFRI. The MLP models are very simple, taking less than 2 minutes to train on a single GPU, since they are $O(1)$ in sequence length after obtaining the mean sequence embeddings. This indicates that the pretrained models have learned rich embeddings of sequence and structure.

Additionally, the finetuned structure model achieves a substantially better AUPRC (0.566 vs 0.446) and max F1 score (0.575 vs 0.460) than DeepFRI. The gap between our structural and non-structural models is much wider than that of DeepFRI, indicating that our model derives a greater benefit from structural information, despite the fact that DeepFRI uses established models for structural processing.

## 4 CONCLUSIONS

In this work we show that standard Transformers are capable of performing structural reasoning by learning an approximately SE(3)-invariant distance filter on attention. We predict that even linearly embedded positions can produce Gaussian attention filters of distance and validate this prediction using experiments on simulated points and proteins. The protein model naturally learns to use the 3D coordinates to measure distance which substantially improves its ability to predict masked tokens. The structural information also materially improves the model's ability to inform function prediction, providing even greater benefit than existing custom-built structural models.

We show that Transformers can learn to measure distance and operate as hybrid structure/language models. In contrast to many conventional structure models which are based on GNNs, Transformers do not explicitly model edges. This admits memory-efficient implementations such as FlashAttention (Dao et al., 2022; Dao, 2023) which allow for fast, fully-connected updates in linear memory. Most structure models store distance in edges which use quadratic memory for fully-connected graphs. Practically, this means that Transformers can perform structural reasoning on more highly connected structures, which may allow them to "see" more while making decisions.

As shown in Section 3.2.2, the pretrained protein model trained with coordinates showed a strong positional dependence in attention in early layers followed by a weak positional dependence in the last few layers. It is possible that this corresponds to the model identifying structural features such as secondary structure and local physics, before encoding these and performing long-range sequential processing. This corresponds with the contemporary trend of preprocessing structural information to create structural tokens for Transformers compared to the more traditional approach of using language model embeddings as input to structural GNNs.

In this work we explore two protein tasks: masked token prediction and function prediction. Virtually all protein learning tasks benefit from combined sequence and structure processing and so this work could be applied across areas including inverse folding, structure prediction, and arbitrary property prediction. As is common in tasks such as inverse folding, the input structures could include more atoms from the backbone. This could be achieved by simply projecting these atom coordinates to the input representation, unlike GNN-based methods which require explicitly including

all pairwise distances in the edge features. Additionally, while proteins are a natural fit for structural Transformers due to their combined sequential and spatial data, there are many other possible applications of this type of model. Some of these include tasks with explicit 3D information such as small molecules and 3D objects. However, there are also tasks where learning an approximate relationship between entities in Euclidean space could help with reasoning, such as vision Transformers (Dosovitskiy et al., 2021) or even large language models.

## 5   REPRODUCIBILITY STATEMENT

Proofs for claims made in Section 2 are available in the Appendix. Code to replicate the experiments is available in an anonymous zipped directory in the supplementary materials. This includes code for models, data download/processing, model training, and evaluation.

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

## A  APPENDIX

### A.1  COORDINATES CAN BE RESCALED FOR BETTER APPROXIMATIONS

The validity of the approximations shown so far depends on the coordinates being small. Figure A1 shows how well $\left(cLN(E_{lin}(\frac{x_1}{c}))\right)\cdot cLN(E_{lin}(\frac{x_2}{c}))$ approximates a quadratic as a function of $c$ and how well the resulting exponential approximates a Gaussian. The scaling parameter $c$ can be learned by the input and output linear maps of the embedding or by the LayerNorm gain parameters. In this way, all coordinates can be rescaled such that the previous sections produce arbitrarily good approximations.

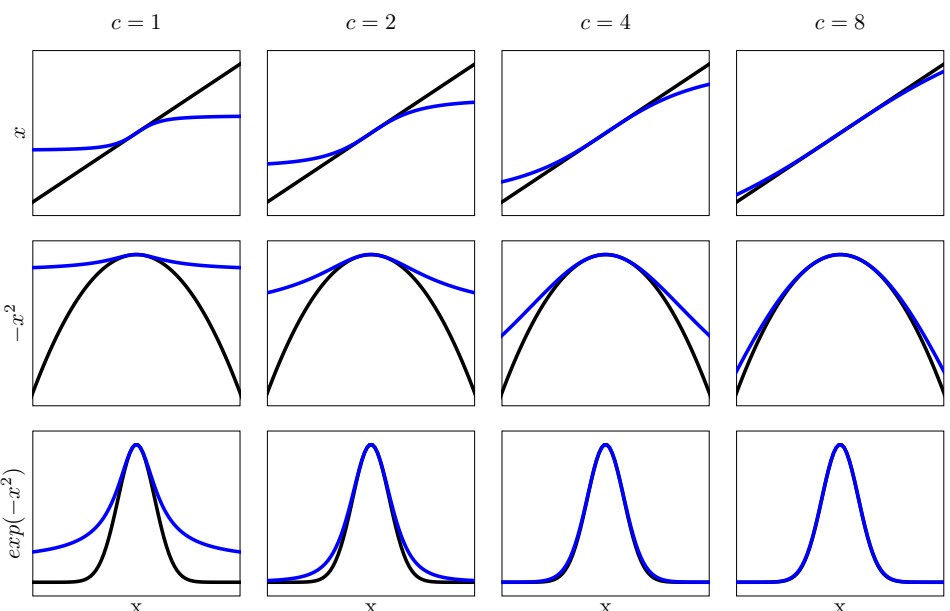

Figure A1: Linear and quadratic approximation and resulting Gaussian approximation of $cLN(E_{lin}(\frac{x}{c}))$ as a function of $c$. Target functions are shown in black and approximations are shown in blue. Increasing the scaling parameter $c$ results in a better approximation without changing the shape of the underlying Gaussian.

### A.2  EMBEDDING PROOFS

#### A.2.1  TRIGONOMETRIC EMBEDDINGS

Consider the embedding $E_{trig}$:

$$E_{trig}(x) = (\cos(x), -\cos(x), \sin(x), -\sin(x)) \qquad (A1)$$

Then the mean, $\mu(E_{trig}(x))$, is:

$$\mu(E_{trig}(x)) = 0 \qquad (A2)$$

and the variance, $\sigma(E_{trig}(x))$, is:

$$\sigma(E_{trig}(x)) = \sqrt{\frac{1}{4}(\sin(x)^2 + (-\sin(x))^2 + \cos(x)^2 + (-\cos(x))^2)}$$

$$= \sqrt{\frac{1}{2}(\sin(x)^2 + \cos(x)^2)} \tag{A3}$$

$$= \frac{1}{\sqrt{2}}$$

So

$$LN(E_{trig}(x)) = \frac{E_{trig}(x) - \mu}{\sigma} = \frac{E_{trig}(x) - 0}{\frac{1}{\sqrt{2}}} = \sqrt{2}E_{trig}(x) \tag{A4}$$

$$LN(E_{trig}(x_1)) \cdot LN(E_{trig}(x_2)) = \sqrt{2}^2 (2\cos(x_1)\cos(x_2) + 2\sin(x_1)\sin(x_2))$$

$$= 4(\cos(x_1 - x_2)) \tag{A5}$$

$$\approx 4 - 2(x_1 - x_2)^2$$

### A.2.2 LAYER NORMALIZATION CAN LEARN APPROXIMATELY QUADRATIC FUNCTIONS OF INPUT

Consider the first-order approximation of $E_{trig}$, $E_{lin}$:

$$E_{lin}(x) = (1, -1, x, -x) \tag{A6}$$

We have

$$\mu(E_{lin}(x)) = 1 - 1 + x - x = 0 \tag{A7}$$

$$\sigma(E_{lin}(x)) = \sqrt{\frac{1}{4}(1^2 + (-1)^2 + x^2 + (-x)^2)}$$

$$= \sqrt{\frac{1}{2}(1 + x^2)}$$

$$\approx \sqrt{\frac{1}{2}(1 + x^2 + \frac{x^4}{4})} \tag{A8}$$

$$= \sqrt{\frac{1}{2}(1 + \frac{x^2}{2})^2}$$

$$= \frac{1}{\sqrt{2}}(1 + \frac{x^2}{2})$$

$$\frac{1 - \mu}{\sigma} = \frac{1}{\frac{1}{\sqrt{2}}(1 + \frac{x^2}{2})} \qquad\qquad \frac{x - \mu}{\sigma} = \frac{x}{\frac{1}{\sqrt{2}}(1 + \frac{x^2}{2})}$$

$$= \sqrt{2}\frac{1 - \frac{x^2}{2}}{(1 + \frac{x^2}{2})(1 - \frac{x^2}{2})} \qquad\qquad\qquad \approx \sqrt{2}(x) \tag{A10}$$

$$\tag{A9}$$

$$= \sqrt{2}\frac{1 - \frac{x^2}{2}}{1 - \frac{x^4}{4}}$$

$$\approx \sqrt{2}(1 - \frac{x^2}{2})$$

So,

$$LN((1, -1, x, -x)) \approx \sqrt{2}((1 - \frac{x^2}{2}), -(1 - \frac{x^2}{2}), x, -x) \tag{A11}$$

In this way, layer normalization can be used to generate approximately quadratic functions of the input. In particular,

$$
\begin{aligned}
LN(E_{lin}(x_1)) \cdot LN(E_{lin}(x_2)) &\approx \sqrt{2}^2 (2(1 - \frac{x_1^2}{2})(1 - \frac{x_2^2}{2}) + 2(x_1 x_2)) \\
&= 4(1 - \frac{x_1^2}{2} - \frac{x_2^2}{2} + \frac{x_1^2 x_2^2}{4} + x_1 x_2) \\
&= 4(\frac{1}{2}(2 - (x_1^2 - 2x_1 x_2 + x_2^2) + \frac{x_1^2 x_2^2}{2})) \\
&= 2(-(x_1 - x_2)^2 + 2 + \frac{x_1^2 x_2^2}{2}) \\
&\approx -2(x_1 - x_2)^2 + 4
\end{aligned}
\tag{A12}
$$

### A.2.3 GATED LINEAR UNITS PROVIDE A BETTER APPROXIMATION

**Lemma A.1.** *ReGLU and SwiGLU can produce functions of $x^2$. In particular:*

$$ReGLU(x) + ReGLU(-x) = SwiGLU(x) + SwiGLU(-x) = x^2$$

*Proof.*

$$
\begin{aligned}
ReGLU(x) + ReGLU(-x) &= max(0, x)x + max(0, -x)x \\
&= max(-x, x)x \\
&= |x|x \\
&= x^2
\end{aligned}
\tag{A13}
$$

Similarly,

$$
\begin{aligned}
SwiGLU(x) + SwiGLU(-x) &= \frac{x^2}{1 + e^{-x}} + \frac{(-x)^2}{1 + e^{-(-x)}} \\
&= \frac{x^2(1 + e^x)}{(1 + e^{-x})(1 + e^x)} + \frac{x^2(1 + e^{-x})}{(1 + e^x)(1 + e^{-x})} \\
&= \frac{x^2(1 + e^x) + x^2(1 + e^{-x})}{1 + e^{-x} + e^x + e^{x-x}} \\
&= \frac{x^2(2 + e^x + e^{-x})}{2 + e^x + e^{-x}} \\
&= x^2
\end{aligned}
\tag{A14}
$$

$\square$

**Theorem A.2.** *With input $\vec{x} = (1, x, x^2)$, there exists a linear embedding, $E$, and a linear map $L$ such that $L(LN(E(\vec{x}))) = (1, x, x^2)/\sigma$ where $1 \leq \sigma \leq 1 + \frac{x^4}{8}$*

*Proof.* Consider the second-order approximation of $E_{trig}$, $E_{quad}(x)$:

$$E_{quad}(x) = (1 - \frac{x^2}{2}, -(1 - \frac{x^2}{2}), x, -x) \tag{A15}$$

We have

$$\mu(E_{quad}(x)) = 0 \tag{A16}$$

$$
\begin{aligned}
\sigma(E_{quad}(x)) &= \sqrt{\frac{1}{4}(x^2 + (-x)^2 + (1 - \frac{x^2}{2})^2 + (-(1 - \frac{x^2}{2}))^2)} \\
&= \sqrt{\frac{1}{4}(2 + \frac{2x^4}{4})} \\
&= \frac{1}{\sqrt{2}}\sqrt{1 + \frac{x^4}{4}} \\
&\approx \frac{1}{\sqrt{2}}\sqrt{1 + \frac{x^4}{4} + \frac{x^8}{64}} \\
&= \frac{1}{\sqrt{2}}\sqrt{(1 + \frac{1}{8}x^4)^2} \\
&= \frac{1}{\sqrt{2}}(1 + \frac{1}{8}x^4)
\end{aligned}
\tag{A17}
$$

$\square$

The error for the LayerNorm-only approximation of $x$ and $x^2$ was $O(n^3)$ and $O(n^4)$ respectively. In comparison, the error given $x^2$ as input for $x$ and $x^2$ is $O(n^5)$ and $O(n^6)$. Thus, simple combinations of ReGLU or SwiGLU layers give us a better approximation of $x$ and $x^2$, which in turn gives us a better approximation of $d^2$. In practice, this may mean that $x$ need not be as small for reasonable approximations to hold which may allow for more stable gradients.

### A.3 RAW ATTENTION PLOTS

In Section 3.2.2, we showed that the attention paid to positional and 3D distance are well-fit by Gaussians. Here, in Figure A2, we provide the same plots but without isolating each of the factors. Note that relative distance, position, and amino acid type are all correlated with one another, especially at linear/3D distance 0.

### A.4 EXTENDED EXPERIMENTS

Here, we report the results of two additional experiments on predicting biological process (Table 2) and molecular function (Table 3) labels, also from the DeepFRI dataset. We were unable to compare these results to DeepFRI because the PDB-only results were not reported. As in the molecular function experiments, the inclusion of structure improved the performance of all models. In these experiments, the performance of the MLP and finetuned Transformers were comparable, and the sequence-only MLP outperformed the sequence-only Transformer for cellular component prediction. It is possible that this is the result of the more expressive Transformer model overfitting to sequence training data, which is then mitigated by the inclusion of structure.

### A.5 MODEL PARAMETER COUNTS

In Table 4, we list the parameter counts for all models used in the paper. The simulated parameter counts will vary slightly as described in the details of each experiment. The pretrained models have the same number of parameters both with and without coordinates. The finetuned models have a slightly higher number of parameters than the pretrained because of the final linear layer which projects to the number of classes. The MLP parameter counts are relatively low because they are conditioned on the (fixed) pretrained embeddings.

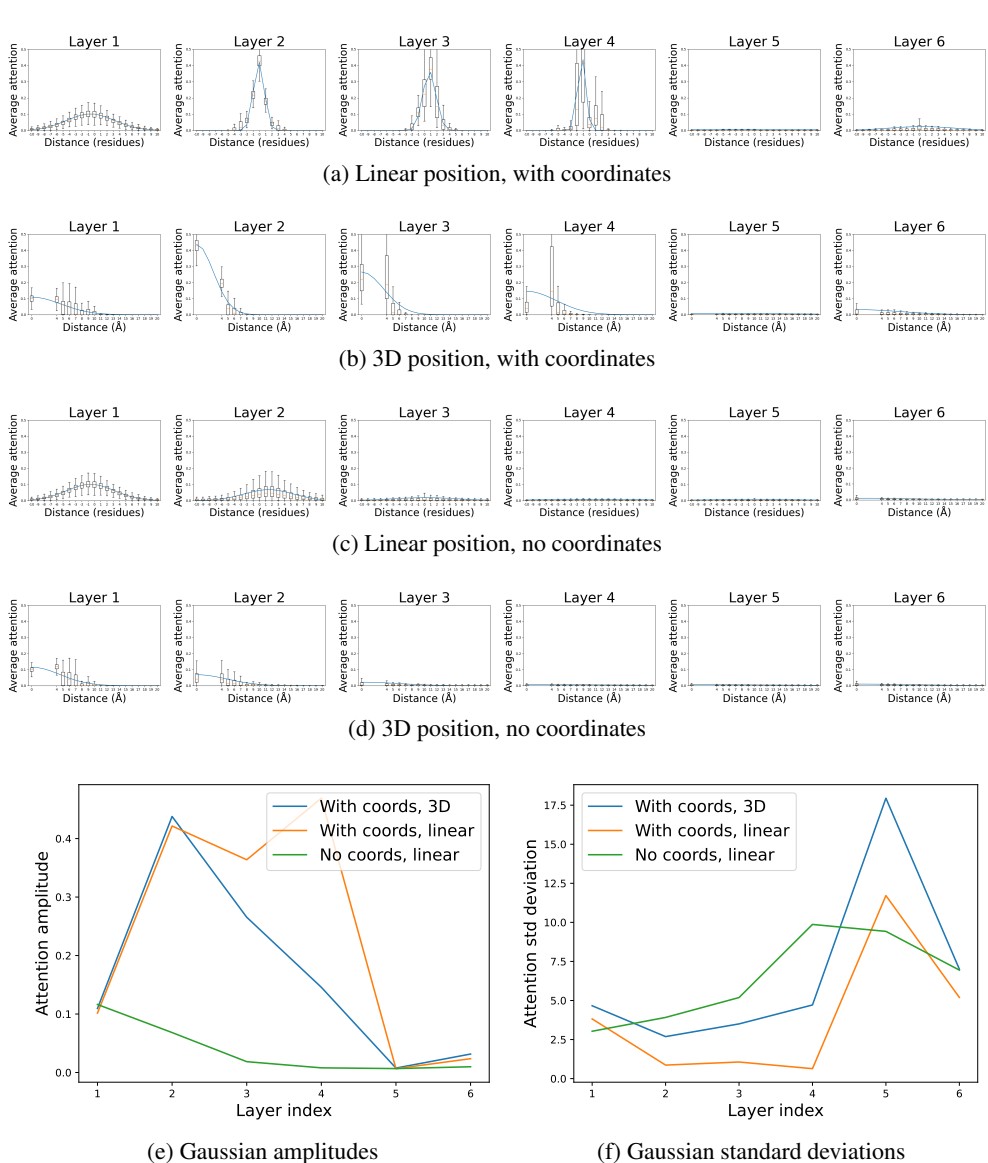

(a) Linear position, with coordinates

(b) 3D position, with coordinates

(c) Linear position, no coordinates

(d) 3D position, no coordinates

(e) Gaussian amplitudes

(f) Gaussian standard deviations

Figure A2: Average attention paid per layer to linear and 3D positional information (a-d) for the unmodified inputs. Fit Gaussian functions are shown in blue for each plot. Amplitudes (e) and standard deviations (f) of the fit Gaussians are shown. The amplitudes are higher and the fits are worse due to the cross-correlation between relative position, 3D distance, and amino acid type.

Table 2: GO biological process prediction results.

| Pretraining (# seqs) | Method | Structure | AUPRC | (Gain from structure) | Max F1 | (Gain from structure) |
|---|---|---|---|---|---|---|
| Ours (∼35K) | MLP | ✗ | 0.197 | | 0.247 | |
| | | ✓ | 0.235 | 0.038 | 0.286 | 0.039 |
| | Finetuned | ✗ | 0.191 | | 0.232 | |
| | | ✓ | 0.244 | 0.053 | 0.280 | 0.048 |

Table 3: GO cellular component prediction results.

| Pretraining (# seqs) | Method | Structure | AUPRC | (Gain from structure) | Max F1 | (Gain from structure) |
|---|---|---|---|---|---|---|
| Ours (∼35K) | MLP | ✗ | 0.281 | | 0.335 | |
| | | ✓ | 0.306 | 0.025 | 0.350 | 0.015 |
| | Finetuned | ✗ | 0.230 | | 0.271 | |
| | | ✓ | 0.308 | 0.078 | 0.343 | 0.062 |

Table 4: Parameter counts for all models.

| Model name | count |
|---|---|
| Simulated model | 1,597,504 |
| Pretrained models | 33,129,242 |
| Finetuned models (cc) | 33,375,322 |
| Finetuned models (mf) | 33,505,283 |
| Finetuned models (bp) | 34,623,409 |
| MLP models (cc) | 2,169,152 |
| MLP models (mf) | 2,342,377 |
| MLP models (bp) | 3,832,727 |

