# OpenReview forum: "Transformers trained on proteins can learn to attend to Euclidean distance"
_ICLR.cc/2025/Conference — Submitted to ICLR 2025_

### Official Review · Reviewer_uMWR · 2024-10-22

**Soundness:** 2
**Presentation:** 2
**Contribution:** 1
**Rating:** 3
**Confidence:** 4

**Summary:**

In contrast with some existing structure prediction/reasoning methods, the authors argue that regular transformers are capable of sophisticated structural reasoning without the assistance of custom invariant graph neural networks. They demonstrate theoretically that hybrid structure-sequence transformers can learn to predict distance matrices. They then evaluate bare-bones transformers on that task as well as protein function prediction, showing parity with baselines.

**Strengths:**

I appreciate the attempt to simplify protein model architectures; these have always been relatively baroque, and so progress toward simpler but performant models would be welcome.

**Weaknesses:**

Overall, I'm unconvinced by the impact of this paper. There are also some question marks about the significance of some of the results. See below for detailed comments:

- > AlphaFold2 (Jumper et al., 2021) and ESMFold (Lin et al., 2022) preprocess protein sequences using Transformers to generate a representation which is used to condition an SE(3)-equivariant structure module to generate protein structures.

I'm going to quibble a bit with this characterization. While it's true that the structures are predicted by a custom structure module, AlphaFold2 also features a *distogram loss* independently of the structure module. While it's true that the Evoformer isn't a bog-standard transformer (it has GNN-flavored triangle attention modules, e.g.), these components were not critically important in the ablation studies for that paper (removing all "triangle" attention corresponded to a dip of < 5 GDT). I'd argue that one of the lessons of AlphaFold2, in comparison to AlphaFold1, was precisely that "(essentially) standard transformers are all you need" for structural reasoning. ESM2, a later development, is a standard transformer, also trained with a distogram loss, and does a pretty good job at contact/distogram prediction, as far as I know. What does this paper add that's not already present there? I think you need to do a much better job separating yourself from prior work. You shouldn't be trying to answer the question "can transformers perform structural reasoning?", since, as I've argued, that's already been established empirically elsewhere. At times you hint at a stronger version of that question: something like "are completely bog-standard transformers *all you need* for structural reasoning?" This is potentially interesting, but smells false to me: while not critically important, the fancy machinery in AlphaFolds 2 and 3 do seem to contribute to those models' edge over bog-standard ESM2. There's also nothing in this paper that would give us reason to believe that ESM2's architecture is not what is holding it back on the structure prediction front.
- How significant is the purported difference in loss values in Figure 2 (a)? Compared to Figure 2 (b), these values are all essentially zero.
- The confidence bars for the "Finetuned" results in Table 5 are absolutely massive---has there been some mistake? If not, there is no sense in which the MLP is "substantially better" than the alternative.
- Some parts of the paper are fairly unnecessary and should be in the appendix: Figure 3, for example, simply shows the benefits of the data augmentation procedure from AlphaFold3 without any modifications. I'm left wondering what is contributed to this study of whether transformers can learn to perform structural reasoning.
- In Table 5, the authors only compare to a baseline from 2021. The authors should add more baselines or do a better job explaining the reasoning behind their choice.
- "The input for these tasks could easily be augmented with more atoms without substantially increasing the memory footprint." - to be convincing, this statement would need to be accompanied by experiments showing that a) *predicted* structures work in this context, since there is no ground truth at the scale at which masked language modeling is typically performed or b) results showing that this sort of structure information can be fine-tuned into an existing PLM pretrained without structure information.

Minor comments (no bearing on score):
- The titles of both panes of Figure 2 are incorrect.
- You should explicitly list parameter counts for all models trained and evaluated in the paper.

**Questions:**

See above

---

> ### Author Response · Authors · 2024-11-21
>
> Thank you very much for your review. We have updated the Prior Work to better contextualize the impact of our work and have addressed the concerns below.
>
> We agree that there are many variants of Transformers which stretch the definition to include any architecture that comprises alternating attention and activation blocks (such as the EvoFormer), but, as you highlighted, our theory and experiments are concerned with standard Transformers. Studying the geometric reasoning capabilities of standard Transformers has three key benefits. We have updated the prior work section to articulate these more clearly and include them here:
>
> (1) They can be implemented in linear memory using methods such as FlashAttention (Dao et al., 2022). This means that they can be trained and run much faster on more standard hardware. If one could show that “standard Transformers are all you need” then one could create a memory-efficient version of AlphaFold which could be run on desktop GPUs.
>
> (2) They are widely used. As you and another reviewer have highlighted, models such as ESM2 (Lin et al., 2023) can learn attention matrices which correlate with structural contacts. This indicates that standard Transformers can learn something about structure, but can they actually formally model structure? This also potentially has implications about the geometric reasoning capabilities of Transformer-based large language models.
>
> (3) An almost-standard Transformer is used as AlphaFold3’s structure module (Abramson et al., 2024). This shows that Transformers should be able to learn to measure distance and potentially learn something analogous to physics. The fact that AlphaFold3 works at all has been the subject of some discussion, and to the best of our knowledge, ours is the first work explaining how this type of Transformer can measure distance to learn a productive form of SE(3)-invariant attention.
>
> By providing a theoretical explanation for how standard Transformers can learn to measure distance, our work improves the understanding of widely used models such as ESM2 and AlphaFold3. It also shows that regular Transformers can be used “out of the box” as memory-efficient structure models.
>
> W1: We have changed the wording to indicate that p=2 is simply the most natural. The loss for the head dimensions is likely much higher because it prevents the model from even representing points in R^n.
>
> W2: It was unclear, but these were not confidence bars. The number in the parentheses was meant to highlight the improvement due to structure. We have split this into a new column to improve clarity.
>
> W3: Figure 3 was partially meant to show the relationship between structural reasoning and SE(3)-invariance. As suggested by another reviewer, we ran an additional experiment to explicitly test for SE(3) deviations and it turns out that the average loss is almost exactly the same as the average deviation from SE(3). We have converted this to a split-panel figure and believe that it now offers a more interesting demonstration of the theory.
>
> W4: We updated this section to include a justification for our comparison. To the best of our knowledge, contemporary, state of the art function prediction methods are all based on pretrained large language models which makes it difficult to compare without retraining a structural model at the scale of large ESM models. Here, DeepFRI is an ideal model to compare to because (1) it was previously state of the art and (2) we could train on exactly the same data which allowed us to isolate the contributions from different architectures and from structure. We believe that a scaled-up version of our model may be able to achieve state of the art in this task but leave this to future work.
>
> W5: We have changed the wording of this statement to highlight that more atoms could be added without substantially increasing memory (as it does in GNNs). It is true that models trained on millions of sequences would require computationally predicted structures, but methods such as ProteinMPNN are trained using backbone atoms from only tens of thousands of structures from the PDB.
>
> Minor points:
>
> We have updated the figure titles for Figure 2 and included a table with model parameter counts in the Appendix.
>
> Dao, Tri, Daniel Y. Fu, Stefano Ermon, Atri Rudra, and Christopher Ré. ‘FlashAttention: Fast and Memory-Efficient Exact Attention with IO-Awareness’. arXiv, 23 June 2022.
>
> Lin, Zeming, Halil Akin, Roshan Rao, Brian Hie, Zhongkai Zhu, Wenting Lu, Nikita Smetanin, et al. ‘Evolutionary-Scale Prediction of Atomic-Level Protein Structure with a Language Model’. Science 379, no. 6637 (17 March 2023): 1123–30.
>
> Abramson, Josh, Jonas Adler, Jack Dunger, Richard Evans, Tim Green, Alexander Pritzel, Olaf Ronneberger, et al. ‘Accurate Structure Prediction of Biomolecular Interactions with AlphaFold 3’. Nature, 8 May 2024.

---

> > ### Comment · Reviewer_uMWR · 2024-11-25
> >
> > I thank the authors for their hard work. I appreciate the updates to the manuscript, but I'm still unconvinced of its impact. As you mention, models like ESM2 and AF3 already give soft, empirical evidence that transformers are all you need for structure prediction. While this paper does offer some harder, theoretical justification for this phenomenon, it does not really demonstrate that "no modifications are necessary for the standard Transformer architecture to learn to perform structural reasoning" in the sense that ESM2 and AF3 are designed in a superfluous way and could have been trained without any of the slight modifications that they include; to make that point convincingly, you'd have needed to train a competitive standard transformer on the task.

---

> > > ### Author Response · Authors · 2024-11-27
> > >
> > > We thank the reviewer for their continued feedback. Through the phrasing “to learn to perform structural reasoning”, we did not intend to imply that all modifications to similar architectures are superfluous. Rather, we are highlighting that Transformers can perform some form of structural reasoning without these modifications. In particular, we show that standard Transformers can explicitly represent 3D coordinates in a way that admits an approximately SE(3)-invariant measure of distance using standard inner product attention. While previous works have shown empirical evidence that Transformers can learn something about structure, our work shows that they can explicitly measure distance, which is an essential prerequisite to more sophisticated structural reasoning.
> > >
> > > Protein structure learning is currently dominated by models which guarantee SE(3)-(in/equi)variance. This bias has a logical basis: the properties of proteins are not intrinsically changed by rotations and translations. However, SE(3) models come with subtle drawbacks such as unstable gradients and high memory usage. Conversely, standard Transformers are widely used for protein sequence analysis and generation. Models such as AlphaFold3 show empirical evidence that the benefits of Transformers can outweigh the drawback of possible SE(3) violations. Still, the question remains:
> > >
> > > “Standard attention is not guaranteed to be SE(3)-invariant but can Transformers learn representations which enable SE(3)-invariant attention?”
> > >
> > > In our manuscript we resolve this question using a new theory for how coordinates can be embedded and normalized which we validate both in simulation and with a realistic pretraining scenario. We also show that the addition of structure allows Transformers to determine protein function more effectively.
> > >
> > > We believe that our work provides a well-supported foundation for the application of standard Transformers to diverse protein learning tasks. We also believe it will guide the rational design of Transformer variants (such as AlphaFold3) which are based on standard inner product attention.

---

### Official Review · Reviewer_ihte · 2024-10-30

**Soundness:** 2
**Presentation:** 3
**Contribution:** 2
**Rating:** 6
**Confidence:** 4

**Summary:**

The paper (1) provides a theoretical explanation for how standard Transformers can learn to measure distance and perform structural reasoning, (2) shows that Transformers indeed learn Gaussian functions of distance and investigate efficient data augmentation methods
which can be used to learn SE(3), and (3) trains a protein masked token prediction model with coordinates and show that finetuning it for function prediction yields a model which outperforms structural GNNs.

**Strengths:**

1. The main idea of a paper is, indeed, interesting. It turns out that to some degree transformers without explicit SE(3) invariance can learn SE(3) invariant functions.
2. Experiment which confirm (1) are provided.

**Weaknesses:**

1. To which degree Equations 2,4 do hold? That is, what is the order of omitted terms?
2. Notation is ambiguous. Sometimes "x" is a vector, sometimes it isn't. For example, A1, A3.
3. I don't understand the purpose of a Section 3.2.3 "PROTEIN FUNCTION PREDICTION".
Using embeddings from pre-trained networks for protein property prediction is an established practice.
4. A very natural experiment is missing. You can take a pre-trained transformer which is presumable SE(3) invariant and shift/rotate coordinates, then check if its output changes.

**Questions:**

1. What is the problem with SE(3)-invariant GNN Transformers? You state that they tend to be memory-intensive, particularly
because attention is performed on edges, which grow as n^2 for fully-connected graphs. But transformers always have n^2 complexity, including yours.
2. Positional encoding in Eq. 5 is not standard. Does your study concern standard trigonometric positional encoding?
3. I don't understand A2. Can you provide a proof?
4. What does max(QKT) mean in Fig. 1 ?

---

> ### Author Response · Authors · 2024-11-21
>
> Thank you very much for your review.
>
> W1: The error in equations 2 and 4 depends on the choice of embedding. For the linear embedding used in the experiments, the post-LayerNorm error of the linear term is O(x^3) small when |x| << 1) and we believe this propagates to O(x^3) for Eq. 2. We discuss this briefly in A.2.3 in comparison to the SwiGLU approximation. Further characterization of the error throughout would be an interesting direction for future study.
>
> W2: Thank you for this feedback. Originally, x could refer to a scalar position whereas x^(->) could refer to a vector position or a shorthand for an embedding. We have removed the embedding shorthand and tidied up all other instances.
>
> W3: We included this experiment to evaluate the downstream utility of a structural Transformer model. We have updated Table 1 to highlight the gain from structure, which shows that adding coordinates to the Transformer substantially improves performance, even more than previous GNN-based structure models.
>
> W4: Thank you for this helpful suggestion. Measuring SE(3)-invariance is tricky because sequence-only models will be trivially SE(3)-invariant and learned distance measures will always have some level of SE(3)-variance. However, in Figure 3b we have incorporated a measure of the divergence between distance measured between randomly rotated structures (translations are always invariant due to recentering) in our augmented and non-augmented models. The adapted figure shows the relationship between SE(3) divergence and validation loss. Both models converge to a loss which is the same as their SE(3) divergence. We still observe that this loss is substantially lower for the model trained on augmented data.
>
> Q1: Transformers always need quadratic compute, and the original implementation of Transformers required quadratic memory, however memory-efficient attention implementations such as FlashAttention (Dao et al., 2022) mean that modern Transformers only require linear memory. This is a huge advantage compared to GNNs because it means we can perform fully-connected attention with reasonable batch sizes on single GPUs. We have added this point to the Prior Work section.
>
> Q2: We appreciate this question from the reviewer. We use standard sinusoidal positional encoding for the linear, sequential positional encoding, but use have a modified form for our 3D positions. The main differences are (1) we include the negatives, which helps with the proof of being unaffected by LayerNorm, and (2) we don’t include other frequencies to simplify our first order, linear embedding. A consequence of our theory is that positional encoding of this form is locally quadratic which means standard positional encoding can learn a sum of quadratic functions locally, which will itself be quadratic. This explains the plots in Figure 4 showing that the linear positions also form approximately Gaussian functions.
>
> Q3: In Eq A2, mu represents the mean which is 0 (cos(x)-cos(x)+sin(x)-sin(x)). We have clarified this in the text.
>
> Q4: As softmax is unaffected by adding a constant to all input values, it is common practice to subtract the max from each value to improve numerical stability. For instance, this is performed in PyTorch and FlashAttention (Dao et al., 2022). This also ensures that each value in the operand of the numerator is <= 0 which means that the unnormalized forms (numerator-only) correspond to Gaussians.
>
> Dao, Tri, Daniel Y. Fu, Stefano Ermon, Atri Rudra, and Christopher Ré. ‘FlashAttention: Fast and Memory-Efficient Exact Attention with IO-Awareness’. arXiv, 23 June 2022.

---

> > ### Author Response · Authors · 2024-11-27
> >
> > We thank the reviewer again for their feedback. As the PDF change deadline is approaching, please do let us know if you have further suggestions for amendments to the manuscript. Otherwise, we are happy to continue the discussion and provide any additional information that may be required to reevaluate the work.

---

> > > ### Comment · Reviewer_ihte · 2024-11-28
> > > **Re: Official Comment by Authors**
> > >
> > > Thank you for clarifications, doing extra experiments and updating the manuscript. I am adjusting my score.

---

### Official Review · Reviewer_HyH3 · 2024-11-04

**Soundness:** 3
**Presentation:** 3
**Contribution:** 2
**Rating:** 5
**Confidence:** 3

**Summary:**

This paper is concerned with analyzing the properties of Transformer-based protein language models, in terms of being able to capture structural properties such as physical distance once proteins are folded.  This idea is supported by theoretical and experimental developments that show that l2 distances (as in Gaussian models) seem to be the natural notion of distance that emerges in appropriate attention mechanisms.  Such Transformer-based protein language models perform well for the downstream task of protein molecular function.

**Strengths:**

Protein language models that operate on amino acid sequences are of significant interest for generative design, among other downstream tasks.  Likewise protein structure prediction, as in the recent Nobel Prize-winning work of AlphaFold*, is of central importance in biochemistry.  This work aims to demonstrate that protein language models themselves have some structural biology capabilities, which also helps with downstream tasks such as protein prediction; this provides significance.  The paper overall is also quite clear in what it does and doesn't do.

**Weaknesses:**

A swath of past work on the interpretability of protein language models, going back to Vig et al. in ICLR 2021 "BERTology meets biology" also show that attention mechanisms capture the folding structure of proteins, connecting amino acids that are far apart in the underlying sequence, but spatially close in the three-dimensional structure (among other results(.  See also references thereto.  It is not clear how much more extra novelty this paper provides beyond this existing line of literature, as no comparison/discussion is made.

Downstream tasks are limited to just one.  Unclear whether the phenomenon of downstream utility is more general than that.

**Questions:**

What is new and exciting, as compared to "BERTology meets biology" and its ilk?

Are there other downstream structural biology tasks that benefit from the structural findings?

---

> ### Author Response · Authors · 2024-11-21
>
> Thank you very much for your review.
>
> W1: Thank you for this suggestion. A similar point was also raised by another reviewer. We have rewritten our Prior Work section to incorporate a comparison to sequence-only models such as Vig et al., 2020. We have also summarized this below:
>
> Our method has some similarities to previous studies such as Vig et al., 2020 and Lin et al., 2023 which show that sequence-only Transformer models trained on proteins learn attention maps which correlate with contact maps. Indeed, both show that Transformers can learn to embed amino acids in such a way that physical proximity can be associated with standard inner-product attention. However, a key difference is that our work shows, both theoretically and practically, that Transformers can explicitly learn an internal representation of 3D structure which admits an approximately SE(3)-invariant measure of distance through attention.
>
> Methods such as Vig et al., 2020 and Lin et al., 2023 may be learning some sort of internal representation of distance similar to that predicted by the theory we present here, despite not receiving explicit coordinates. This is a plausible explanation for the phenomena observed in previous works and would fit well with the observation that language models such as ESM are useful for conditioning structure prediction models. However, to the best of our knowledge, our work is the first to provide evidence for the fact that Transformers can formally model structure as 3D coordinates, rather than just learning sequence patterns which correlate with structure.
>
> W2: Thank you for this suggestion. To complement the molecular function experiment, we have added two experiments to predict biological process and cellular component, which we have added to the Appendix (A.4). We were unable to compare the performance to DeepFRI because the PDB model results were not reported for these labels, but we observed a similar phenomenon for our experiments where adding coordinates improved all models.
>
> Vig, Jesse, Ali Madani, Lav R. Varshney, Caiming Xiong, Richard Socher, and Nazneen Rajani. ‘BERTology Meets Biology: Interpreting Attention in Protein Language Models’, 2020.
>
> Lin, Zeming, Halil Akin, Roshan Rao, Brian Hie, Zhongkai Zhu, Wenting Lu, Nikita Smetanin, et al. ‘Evolutionary-Scale Prediction of Atomic-Level Protein Structure with a Language Model’. Science 379, no. 6637 (17 March 2023): 1123–30.

---

> > ### Author Response · Authors · 2024-11-27
> >
> > We thank the reviewer again for their feedback. As the PDF change deadline is approaching, please do let us know if you have further suggestions for amendments to the manuscript. Otherwise, we are happy to continue the discussion and provide any additional information that may be required to reevaluate the work.

---

### Official Review · Reviewer_WYJW · 2024-11-08

**Soundness:** 3
**Presentation:** 3
**Contribution:** 3
**Rating:** 8
**Confidence:** 3

**Summary:**

This paper explores the ability of Transformers to attend to spatial structures without relying on explicit structural modules.

By feeding linear embeddings of coordinates into Transformers, the Authors demonstrate that these models can approximate Gaussian spatial attention, enabling them to estimate and make use of spatial relationships.

The Authors validate this approach in a sequence of steps, from simplified models to a protein language model similar to ESM1, concluding that a structurally enriched Transformer outperforms traditional graph neural networks (GNNs) in function prediction.

This study contributes to using Transformers in spatial tasks, illustrating a new and useful capability of these models.

**Strengths:**

This paper seems to bring an original contribution, by demonstrating that Transformers can handle 3D structural reasoning on their own.
This capability challenges the reliance on dedicated modules for spatial tasks that enforce symmetries and equivariance and introduces a new approach to protein structure modeling and other applications involving 3D data.

The Authors provide a sound theoretical explanation of how Transformers can approximate Gaussian distance filters making use of coordinates embeddings, enabling them to encode distance relationships with the attention mechanism. In particular, it is appreciable their effort to make the findings accessible through the discussion of simple cases, providing a clear motivation for the approach.

The writing is great and makes the reading very easy and pleasant.

**Weaknesses:**

The quality of some of the plots is not excellent. For example, in Fig. 4 a-d plots are not very readable (tick labels are essentially invisible), and maybe a different strategy to convey the relevant information could be implemented.

**Questions:**

I did not get this comment "The version with coordinates also had a lower validation loss for the same training loss, and so the structural features learned early in training may be more robust to dissimilarity in sequence space."; it seems to me, from the plot that the two models never reach the same training loss. It may be trivial but can you clarify this passage please?

I suggest to update the references section because some of the papers cited have been published in the meantime (e.g, "Evolutionary-scale prediction of atomic level protein structure with a language model", by Lin et al.).

**Details Of Ethics Concerns:**

No ethics concerns

---

> ### Author Response · Authors · 2024-11-21
>
> Thank you very much for your positive review.
>
> W1: There was an error in the original figure where the first panel contained a 3D distance plot rather than a residue distance plot. We hope it is more clear now. Our preference is to keep each of the panels because it helps to illustrate (1) that the attention plots are well-approximated by Gaussians and (2) that the nature of the attention changes layer-by-layer. If you think it would help, we can reduce the number of ticks which would allow us to increase the tick label size.
>
> Q1: The phrasing here was unclear and we have updated it. The point is that the structural version reaches parity on training loss early (after only 8 epochs). At this point the validation loss is lower than the non-structural version, so the model has learned to achieve that training loss using structural features which seem to generalize better.
>
> Q2: Thank you for pointing this out. We have updated the ESM2 reference.

---

> > ### Author Response · Authors · 2024-11-27
> >
> > We thank the reviewer again for their feedback. As the PDF change deadline is approaching, please do let us know if you have further suggestions for amendments to the manuscript. Otherwise, we are happy to continue the discussion and provide any additional information that may be required to reevaluate the work.

---

### Meta-Review · Area_Chair_GXei · 2024-12-20

**Metareview:**

The paper provides a theoretical analysis demonstrating how Transformers are capable of performing structural reasoning, suggesting that they can learn Gaussian functions related to distances. To validate these theoretical claims, the authors trained a protein language model with structural enhancements. The experimental findings indicate that the structurally enhanced Transformer surpasses competing baselines in the task of protein function prediction.

The paper presents a theoretical perspective indicating that Transformers are equipped to tackle tasks involving 3D structural reasoning and includes experiments to bolster these theoretical claims. Nonetheless, the evidence presented falls short of conclusively proving that the standard Transformer architecture is adept at structural reasoning. After thorough consideration of other submissions in the same batch, I find myself recommending rejection for this paper. I encourage the authors to refine their work and consider resubmission to a future conference or journal.

**Additional Comments On Reviewer Discussion:**

## Points Raised by Reviewers:
1. Impact and Novelty (reviewer uMWR and HyH3): Skeptical about the paper's impact and the novelty of the results, questioning the significance of the presented loss values and the large confidence intervals reported.
2. Characterization of Existing Methods (reviewer uMWR): Challenges the paper's description of AlphaFold2 and ESMFold, suggesting that the capabilities of standard transformers for structural reasoning have already been established.
3. Choice of Baselines and Additional Experiments (reviewer uMWR, ihte, HyH3): Recommends adding more baselines and experiments or justifying the choice of the current baseline.
4. Assumptions on Memory Footprint (reviewer uMWR): Questions the claim that more atoms could be added to the input without significantly increasing memory usage.

## Author Responses:
1. Clarifications and Amendments: The authors update the Prior Work section to better contextualize their work and differentiate it from previous studies. They assert that their work provides theoretical and practical evidence that Transformers can learn an internal representation of 3D structure, which is novel compared to sequence-based attention correlations demonstrated in prior work.
2. Additional Experiments: In response to concerns about generalizability, the authors add experiments on predicting biological processes and cellular components, reporting improved model performance with the addition of coordinates.

## Weighting Points in the Final Decision:
While the authors addressed some concerns, the novelty issue remains. The paper should provide compelling evidence to substantiate its results and unique contributions.

---

### Decision · Program_Chairs · 2025-01-22

Reject